# Health Disparities Research Framework Adaptation to Reflect Puerto Rico’s Socio-Cultural Context

**DOI:** 10.3390/ijerph17228544

**Published:** 2020-11-18

**Authors:** Irene Lafarga Previdi, Carmen M. Vélez Vega

**Affiliations:** Center for Collaborative Research in Health Disparities, UPR Medical Sciences Campus, San Juan 00936, Puerto Rico; carmen.velez2@upr.edu

**Keywords:** health disparities, social determinants of health, socio-ecological framework, Puerto Rico

## Abstract

In this article we aim to briefly describe how Puerto Rico’s living conditions influence adverse health outcomes at an individual, community and population level using the National Institute of Minority Health and Health Disparities (NIMHD) Research Framework that considers multiple factors and their intersecting influence. People living in Puerto Rico face significant levels of poverty, a deficient infrastructure, a fragile healthcare system and the continuing dismantling of the public education system as well as hazardous environmental exposures. The treatment of Puerto Ricans as second-class citizens due to the federal policies of the U.S. government and also the mismanagement of funds from local authorities impacts the prevalence of chronic health conditions and vulnerability to disasters such as hurricanes, earthquakes and pandemics. Puerto Rico’s health disparities are rooted in historical, cultural, political and economic factors that have an impact on biology, interpersonal and environmental aspects. In order to significantly reduce health disparities, systemic change is needed at a local, national and federal level. Interventions must consider how social determinants impact the quality of life and seek to impact the intersections of different contexts that have an effect at an individual, interpersonal, communal and societal level. This can be achieved through evidence-based, culturally appropriate and community based as well as translational research approaches that seek to impact behavior and social economic factors.

## 1. Introduction

Puerto Rico is an archipelago composed of Isla Grande, Vieques, Culebra and other smaller islands located in the Caribbean with a population of 3.2 million people [1]. Puerto Rico has been a non-incorporated territory of the United States since 1898 like the U.S. Virgin Islands, Guam and Samoa. Puerto Ricans have been U.S. citizens since 1917 but are still unable to vote in presidential elections and only have one non-voting representative in Congress. Since 1920, with the Merchant Marine Act (more popularly known as the Jones Act), the maritime waters and ports of Puerto Rico have been controlled by U.S. agencies, which has led to an increase in the cost of food and other supplies, which can be higher than in the U.S. [2]. During the 20th century, United States based and other foreign manufacturers of items such as textiles, electronics and pharmaceuticals came to the islands to establish factories due to significant tax incentives. However, after changes in public policy most of these industries left, which led to major layoffs and the presence of significant environmental contaminants [3]. In addition, the U.S. Navy occupied the islands of Culebra and Vieques to use the territories as a bombing range and a site for military training exercises during the second half of the 20th century [4].

These living conditions have led to adverse health outcomes that impact individuals and their families as well as the general population. In general, Puerto Rico has several health disparities in relation to several chronic conditions such as diabetes, asthma, cancer, HIV and cardiovascular diseases [5]. When compared with the general population of the United States, the overall occurrence and mortality rates are higher among people living in Puerto Rico [5]. In this article we aim to briefly describe how Puerto Rico’s living conditions influence adverse health outcomes at an individual, community and population level using the National Institute of Minority Health and Health Disparities (NIMHD) Research Framework, which considers multiple factors and their intersecting influence. This framework is related to the concept of Social Determinants of Health, which focuses on how the conditions in which people are born, live, work, learn and age impact their quality of life [6]. Social determinants of health refer to the influence of the social, cultural, economic and political environment on individual and population health. In Puerto Rico, while a small territory, there is great diversity in living conditions according to socio-economic class, race, gender, gender identity and sexual orientation. For the purpose of this article we will focus on the overall general population and provide a broad description of how the domains of influence intersect with the levels of influence that produce health outcomes that result in health disparities. However, we must mention that one of the limitations of this analysis has been the lack of updated statistics, which is a reflection of the state of available data in Puerto Rico and/or about the Puerto Rican population.

## 2. Living Conditions in Puerto Rico

### 2.1. Social and Economic Conditions

During the mid-20th century, the economic development plan popularly known as Operation Bootstrap helped to transform the local economy from an agrarian society to an industrialized one at an accelerated rate. The plan relied on incentives such as federal tax exemptions and low labor costs to attract U.S. companies to the archipelago [7]. It also created a shift towards an export-based economy in which the production of materials and services was aimed at the U.S. market instead of investing in promoting the local economy [7] According to the Center for a New Economy, “between 1950 and 1980, per capita gross national product grew nearly tenfold in Puerto Rico, and disposable income and educational attainment rose sharply” [8]. However, when Section 936 was repealed in 2006, which had allowed U.S. businesses to operate without paying taxes, it caused many manufacturers to leave Puerto Rico, which left many people without jobs. Due to this, the 2008 global financial crisis hit the archipelago especially hard, which led the government to implement austerity measures such as massive layoffs of public workers and other budget cuts [7].

During the last decade there have been significant incremental budget cuts in education, health, culture and environmental protections. Puerto Rico is presently undergoing a socio-economic crisis because of an enormous government debt that has led to the institution of a Fiscal Oversight and Management Board appointed by the Federal Government, which has led to severe austerity measures and emigration [9]. It is worth mentioning that this debt was partly due to the fact that investors in local municipal bonds received favorable tax treatment and they took advantage of this [10]. This favorable treatment consisted of giving bond investors higher returns and loosening borrowing limits; lenders to Puerto Rico were exempt from local, state and federal taxes [7]. In addition, the Puerto Rican constitution allowed the government to balance its budget with debt and, after the repeal of the Internal Revenue Code Section 936 that allowed American businesses to operate tax-free in Puerto Rico, the government began to rely more on borrowed funds from bond issuance to balance its budget [7,10].

Currently, 43.1% of the population lives under poverty conditions and in 2017 57.8% of children lived in poverty [1]. In 2017, the median household income was USD 19,775, which was approximately half of the median household income of Mississippi (USD 42,009), the poorest state of the United States, and approximately one third of the median household income in the United States (USD 57,652) [5]. During this same year, probably influenced by the aftermath of Hurricanes Irma and María, 97,000 people migrated to the United States while 20,000 returned to live in Puerto Rico [11]. In February 2020, just before the COVID-19 pandemic, unemployment in Puerto Rico was at 8.8% [12]. A recent study found 33.2% of the population aged 18 and older presented food insecurity [13]. The study also reported 44.3% of people who were food insecure perceived their health to be average or poor. In Puerto Rico, there are currently more than two hundred hazardous waste sites with twenty-two of them being on the National Priorities List and 17 Superfund sites (EPA Superfund Program) [3]. 

### 2.2. Infrastructure

Infrastructure is central to the quality of life, productivity and economic development of individuals and communities. However, Puerto Rico relies on a deficient infrastructure that negatively impacts the daily life of residents and also hinders social and economic development. According to the American Society of Civil Engineers’ 2019 Report Card, the island has an overall grade of D. This is because we lack programmed funding to invest in the comprehensive maintenance of the existing infrastructure as well as exploring modern and more sustainable options [14]. Currently, an estimated 40–60% of storage capacity in potable water is lost due to sedimentation build up and an estimated 58% of non-revenue water is lost as a result of leaky pipes and tank overflows. This results in constant water rationing in spite of significant annual rainfall as well as the frequent interruption of services. In addition, there is the presence of harmful chemicals and substances in the water supply that negatively affects health [15]. Before Hurricanes Irma and María hit Puerto Rico in 2017, the island’s energy infrastructure was deficient because of a lack of proper maintenance. Puerto Rico also lacked source diversity despite being a tropical island with constant sunlight with approximately 98% of electricity generated by fossil fuels. After the devastation caused by the storms, efforts were focused on the short-term goal of restoring power instead of developing a more sustainable and resilient energy system. This has resulted in frequent blackouts, the interruption of services and a continued increase in prices. In addition, landfills in Puerto Rico are often unregulated or operate without updated permits, which results in non-compliance with Environmental Protection Agency standards. Capacity is a also major issue, which has been exacerbated by the debris from the 2017 hurricane season.

### 2.3. Healthcare System

There are several structural challenges regarding the how the healthcare system operates and its capacity to address the population’s health needs. In the Puerto Rico Health Care Infrastructure Assessment: Site Visit Report from 2017 four main challenges were identified: (1) the privatization of the public health care system and the rise of managed care, (2) the aging of the Puerto Rican population combined with high rates of poverty and some chronic health conditions, (3) the economic instability and low private sector tax base of the commonwealth and (4) the high cost of living in Puerto Rico, poorly coordinated care, difficulty receiving referrals and long wait times [16] (p. 3). 

In addition, according to this Report, which was based on the experts that were interviewed, the current structure of the health care system has made it difficult to have high-quality medical services. Most Puerto Ricans depend on public health insurance such as Medicaid and Medicare; commercial insurance or employer-provided insurance sectors are significantly smaller. The payment rates of Medicaid managed care and Medicare Advantage in Puerto Rico are less than 40% of the average rates for both programs in the U.S. [16]. 

In the last ten years, 5000 doctors have left Puerto Rico and have gone to work in the United States for economic reasons. The decline in the population of doctors is 36% compared with the decline of the general population, which is 9% [17]. This means that there are fewer specialists on the island and that patients have to wait months for an appointment or are also forced to emigrate to seek better medical services. According to the Association of American Medical Colleges, in 2018 there were 306.3 physicians per 100,000 people and a total of 9787 active physicians to attend 3.2 million people living in Puerto Rico [18]. The proposed budget by the Fiscal Oversight and Management Board considered an increase of USD 242.9 million to the Department of Health but other institutions like the Health Insurance Administration suffered a cut of USD 586.3 million and the Comprehensive Cancer Center a reduction of USD 4.3 million [19]. Basic healthcare services in Puerto Rico also face major risks because of an expected funding reduction of the Medicaid Program, which could affect healthcare access for 1.5 million people. Those at risk are Puerto Rico’s most vulnerable populations; low income families and children, pregnant women, the elderly and people with disabilities [5].

### 2.4. Educational System

According to Onieva López [20], the main problems of the Puerto Rican educational system are:

“The inconsistency in implementing the educational plans of the different governments; the existence of an obsolete curriculum that has failed to implement new technological resources in classrooms; that teachers are overloaded, poorly prepared, unmotivated, and poorly resourced; blaming teachers for the ills of the country’s education; school absenteeism; violence in classrooms; poor infrastructure of schools and their poor maintenance; fast schools; and the continuous attack on the public school and its devaluation, while important areas of the educational system are privatized”[20] (pp. 69–70).

The recent closure of public schools carried out by the former Secretary of Education Julia Keleher, who presently faces federal charges for corruption during her charge, had a major impact on rural areas on the island where 65% of schools were closed compared with 35% in urban areas according to research from the Center for Puerto Rican Studies at Hunter College in New York. The analysis established that 265 schools were closed [21]. However, a local newspaper revealed that 438 schools were closed in the previous two years [22]. 

According to data provided by the Department of Education in the fiscal year 2017–2018, 183 schools were eliminated while in fiscal year 2018–2019 there were 255 [22]. According to a report by UC Berkeley’s Othering and Belonging Institute and San Juan-based Centro para la Reconstrucción del Hábitat, the school closures generated meager financial gains and were also ineffective in saving funds. The researchers reviewed 123 contracts related to the closed schools and found that only 8% had been sold and the rest rented, generating in total less than USD 4.3 million, which is much less than government initially expressed [23]. Meanwhile, earlier this year it was reported that the University of Puerto Rico, the only public institution of higher education with 11 campuses around the island, could be in jeopardy if the USD 71 million contemplated for the next fiscal year is cut. Added to the USD 331 million of prior reductions, the University would have USD 400 million less in its budget [24].

## 3. Health Disparities in Puerto Rico

Health disparities are differences of health based on the overall rate of disease incidence, prevalence, morbidity and mortality that affect certain populations [25]. Ethnic and racial minorities, the LGBTTIQ population, population with disabilities, low income populations, immigrants and refugees are considered health disparity populations because they are disproportionally affected by chronic diseases when compared with the general population. In Puerto Rico, while a small territory, there is a great diversity in living conditions according to socio-economic class, race, gender, gender identity and sexual orientation. However, for the purpose of this article we will focus on the overall general population and provide a broad description of how the domains of influence intersect with the levels of influence that produce health outcomes that result in health disparities. Puerto Rico has several health disparities in comparison with the general population in the United States. We have greater prevalence and mortality rates of several chronic conditions such as diabetes, asthma and cardiovascular diseases [9]. The percent of adults reporting fair or poor health in 2017 reached 37.1% in Puerto Rico compared with 25.3% in Mississippi and 18.4% in the general U.S. population. The healthiest jurisdiction according to this indicator was the District of Columbia, reporting 10.8%. Diabetes prevalence in Puerto Rico in 2017 was 17.2% compared with 10.5% in the U.S. Asthma and high blood pressure prevalence was 12.2% and 44.7% in Puerto Rico, respectively, compared with 9.4% and 32.3% in the U.S., respectively [9].

In 2017, Puerto Rico also had a rate of 11.4% of preterm births, which ranks among the highest rates of preterm birth at an international level [26,27]. For HIV, in 2018 for the United States and six dependent areas, the overall rate of cases was 11.5; meanwhile in Puerto Rico it was 15.6 [28]. Puerto Rico was also the focal point of the Zika virus epidemic [29]. Over the course of a year and four months, more than 35,400 cases were confirmed in the Archipelago. This represented 85% of all cases reported in the U.S. and its territories. Another relevant health disparity is the environmental exposure to contaminants such as phthalates. In a study by Puerto Rico Testsite for Exploring Contamination Threats (PROTECT) of a cohort of over 2000 pregnant women from the northeastern Karst region of the island found that compared with women in the general U.S. population, urinary concentrations of metabolites of di-n-butyl phthalate (DBP) and di-isobutyl phthalate (DiBP) were higher among Puerto Rican pregnant women [30]. Meanwhile, in the municipal island of Vieques, which was used as target practice by the U.S. Navy, cancer rates are higher than those in any other municipality in Puerto Rico [4]. The U.S. Navy has admitted to using heavy metals and toxic chemicals but has continuously denied any link between these substances and the health conditions of Viequenses. When comparing cancer rates between the Puerto Rican population and the rest of the U.S. we find several differences. For example, Puerto Rico’s incidence rates for infection related cancers are higher than those in non-Hispanic whites [30]. There are also disparities in mortality rates. Breast cancer is the leading cancer cause of death among women and prostate cancer is the leading cancer cause of death for men, in contrast to the U.S. where lung cancer is the leading cause of cancer death [31].

## 4. National Institute of Minority Health and Health Disparities (NIMHD) Framework Considering the Puerto Rican Population

The National Institute of Minority Health and Health Disparities (NIMHD) Research Framework provides an insight into the multiple intersections that influence health outcomes in minority populations. According to Alvidrez and collaborators [25], the NIMHD Research Framework is based on the National Institute of Aging (NIA) Research Framework and the Socio-Ecological Model. The NIA Research Framework considers different factors (biological, behavioral, socio-cultural and environmental) that influence health outcomes across the lifespan [32]. Meanwhile, the Socio-Ecological Model considers the complex bi-directional relationship between individual, relationship, community and societal factors [33]. This theory was initially proposed by Bronfenbrenner and considers different systems such as the microsystem (interactions and relationships of the immediate surroundings), the mesosystem (work, school, church and neighborhood contexts), the exosystem (community contexts and social networks), the macrosystem (societal, religious and cultural values and influences) and the chronosystem (internal and external elements of time and historical content and the influence of policy) [34]. 

We also consider that this framework is related to the concept of Social Determinants of Health, which focuses on how the conditions in which people are born, live, work, learn and age impact their quality of life [6]. This refers to the impact of structural systems (which can be immersed in racism, sexism, LGBTTIQ phobia, ableism and classism) in the development of communities and the individuals who inhabit them. Resources that can improve the quality of life can have a significant influence on population health outcomes. A few examples of these resources include safe and affordable housing, access to quality education, public safety, availability of healthy foods, access to quality health services and environments free of life-threatening toxins. A “place based” organizing framework identifies five key areas of social determinants of health that must be addressed in order to reduce health disparities: (1) economic stability, (2) education, (3) social and community context, (4) health and health care, (5) neighborhood and built environment [6]. 

Health disparities do not occur in a vacuum, which is why the concept of Social Determinants of Health (SDH) along with the previously mentioned Socio-Ecological Framework provide a holistic and integrative approach where socio-economic policies and also cultural patterns are taken into consideration when studying differential health outcomes in minority populations. In considering how living conditions impact health outcomes, we can infer that adverse experiences have a negative impact on the wellbeing of communities as well as individuals. There is growing evidence that an exposure to adverse living conditions can have a negative impact on childhood development, which can lead to disease later in life [35]. We must also take into account that living conditions are influenced by systemic structures that have an effect on the social, physical and natural environmental conditions in which populations exist. Puerto Rico’s colonial relationship with the U.S., current economic crisis and recent emergencies due to natural disasters have had an impact at a community, interpersonal and individual level. As Thomas Frieden’s five-tier Health Impact Pyramid suggests, “the greatest health impact likely will come from interventions targeting socio-economic factors that drive health disparities across multiple conditions” [35] (p. 1417). 

Here are a few federal policies that have impacted the quality of life for Puerto Ricans:-The Jones–Shafroth Act (1917): through this, Puerto Ricans were granted full U.S. citizenship and it established an elected (rather than presidentially-appointed) legislature on the island; however, this did not mean that they could vote in presidential elections or have voting representatives in Congress. The case Balzac v. Porto Rico (1922) established: “the granting of citizenship to Puerto Ricans did not mean that Congress had expressed an intention of eventually incorporating Puerto Rico as a state...the Act merely allowed the residents of Puerto Rico free entry into the United States, where they could exercise full rights at citizens.” Balzac v. Porto Rico firmly cemented Puerto Ricans’ status as second-class citizens and served to keep the political status of the island in limbo indefinitely by assuring that no promises regarding statehood and equality were made [36] (p. 3).-The Jones Act (Marine Merchant Act, 1920): this legislation was related to protecting U.S. interests in relation to national shipping and interstate commerce. After World War II, it established cabotage regulations that required that only U.S. ships could transport materials and supplies from one domestic port to another [36]. This meant that: *“basic shipments of goods from the island to the US mainland, and vice versa, must be conducted via expensive protected ships rather than exposing them to global competition. That makes everything Puerto Ricans buy unnecessarily expensive relative to goods purchased on either the US mainland or other Caribbean islands, and drives up the cost of living on the island overall [37].”* This increase in prices, along with a minimum wage of USD 7.25 and an increase in part time jobs without marginal benefits, had an impact in access to health services and medications, nutritious food, secure housing and educational opportunities.-The Puerto Rico Oversight, Management, and Economic Stability Act (PROMESA, 2016): this law was created to supervise the government efforts to manage the payment of bonds related to the accumulated debt. This legislation has contributed to refute the discourse that Puerto Rico is a self-governing territory [38]. The Harvard Law Review study of PROMESA concluded “the Board can influence nearly any area of policy making in Puerto Rico. The Board also undercuts any autonomy Puerto Rico had in respect of economic and social affairs. The extensive powers conferred on the Board are fundamentally incompatible with U.S. standards for self-government” [39]. The Board, whose members were appointed by the U.S. Congress, has prioritized the debt payment over the funding of social support programs, contributing to an increase in poverty in already underserved populations.

Another legal aspect that impacts the quality of life of Puerto Ricans is federal transfer payments. These are divided into two categories, earned benefits such as Medicaid, Social Security and veterans’ pensions and direct aid such as nutritional assistance and grants [36]. In 2019, most of the transfer payments to individuals were earned benefits and not welfare related [40]. However, federal transfers to individuals in Puerto Rico are arbitrarily capped at levels below those of the states in U.S. [36]. This does not take into account the population levels or the need for these kinds of services but instead is a form of selective discrimination based on the fact that the archipelago is an unincorporated territory and not a state. If Puerto Rico were treated as a state, federal transfer programs for health, i.e., Medicaid and nutrition, i.e., Women, Infants and Children (WIC) Nutrition Program would be substantially greater. However, the fact is that Puerto Rico receives fewer U.S. federal dollars to assist with social programs than states with comparable populations [10]. This has, as a consequence, meant that people living in Puerto Rico have limited access to social support measures that could help to improve living conditions and thus their overall health.

Puerto Ricans are being treated as second-class citizens due to the unequal treatment of the U.S. government through federal regulations, which are implemented arbitrarily without any consultation or participation from Puerto Ricans. These federal regulations influence the living conditions of Puerto Ricans at individual, interpersonal, community and societal levels (see Table 1). This has an impact on health outcomes and the quality of life, for example, higher costs for basic needs including healthcare and food due to cabotage regulations. This also can be understood through the concept of embodiment, which refers to “how we, like any living organism, literally incorporate, biologically, the world in which we live, including our societal and ecological circumstances” [41] (p. 351). This construct takes into consideration the fact that humans are social beings as well as biological beings, which means that in order to understand health disparities we must understand the living conditions of individuals and communities. For example, factors such as food insecurity and food deserts, deficient infrastructure (i.e., electricity, potable water, public transportation, internet access), economic and social deprivation, toxic environmental exposures, lack of access to adequate healthcare and poor job conditions all leave a physical mark on the body of individuals, communities and societies. This tells us that “it is no accident that from population patterns of health, disease, and wellbeing it is possible to discern the contours and distribution of power, property, and technology within and across nations, over time” [41] (p. 350).

The NIMHD Research Framework (see Table 1) serves to illustrate how the living conditions in Puerto Rico previously mentioned (social and economic, infrastructure, healthcare system, educational system, political environment) influence the quality of life of the general population in the archipelago. This can help us to understand the various health disparities that occur among Puerto Ricans, which are the product of multiple intersections between individual, relationship, community and societal factors. When studying adverse health outcomes, the various levels of domains of influence should be considered (biological, behavioral, physical environment, socio-cultural environment, healthcare system) whether the focus in on an individual case, a particular community or the general population. This is because health is the result of not only our biological constitution but also our behaviors, which are shaped by societal norms, policies and laws, the natural environment, access to resources and even historical events. In this article we presented a brief overview of several social determinants of health that affect people living in Puerto Rico to give an idea of how health disparities are an embodiment of our circumstances.

## 5. Conclusions

Puerto Rico’s health disparities are rooted in historical, cultural, political and economic factors that have an impact on biology, interpersonal and environmental aspects. The NIMHD Research Framework helps illustrate the complex relationship between the environment and the individual and how health disparities may originate. This analytical instrument can serve to visualize the multitude of factors that impact on the health outcomes of populations living in poverty. With almost half the population living in poverty, significant budgets cuts in education and healthcare, a deficient infrastructure, an economic crisis and a very limited political autonomy, Puerto Rico is very vulnerable to significant health disparities with regard to chronic health conditions and disasters such as hurricanes, earthquakes and pandemics. Being resilient is not enough to significantly reduce these health disparities and a systemic change at a local and federal level is needed. In order to mitigate health disparities and adverse health outcomes, we must implement evidence-based strategies, multilevel interventions, participate in public policy development and implementation and promote community engagement in translational research processes.

## Figures and Tables

**Table 1 ijerph-17-08544-t001:** National Institute of Minority Health and Health Disparities (NIMHD) Research Framework adapted to reflect historic and socio-cultural influences of the Puerto Rican population.

Domains of Influence	Levels of Influence			
	Individual	Interpersonal	Community	Societal
**Biological**	*Biological Vulnerability and Mechanisms*	*Caregiver-Child Interaction* *Family Microbiome*	*Community Illness Exposure* U.S. Navy presence in Vieques and Culebra Ashes from AES Power Plant Hazardous waste sites *Herd Immunity*	*Sanitation* Water quality *Immunization* 85% in 35 mo in 2014 *Pathogen Exposure* Dengue, Chikungunya, Zika, Leptospirosis, STI
**Behavioral**	*Health Behaviors* *Coping Strategies* Resilience Religion/Spirituality Communal bonds	*Family Functioning* Extended family Women heads of household *School/Work Functioning*	*Community Functioning* Solidarity Community Councils	*Policies and Laws* Puerto Rico Oversight, Management, and Economic Stability Act (PROMESA) Law Jones Act Civil Code
**Physical/Built Environment**	*Personal Environment*	*Household Environment* Deficient infrastructure Public housing Closed neighborhoods *School/Work Environment* School closures Deficient infrastructure	*Community Environment* Natural resources Social capital *Community Resources* Tourism Local businesses Local non-governmental organitzations	*Societal Structure* Almost 50% live in poverty Diaspora in U.S.
**Socio-Cultural Environment**	*Socio-demographics* 20% less than 18 years 20% 65 years or more 60% 18–64 years old *Limited English* English taught as a second language *Cultural Identity* Boricua/American citizens *Response to Discrimination* Historical trauma Colonized mindset	*Social Networks* Extended family Organized Communities *Family/Peer Norms* *Interpersonal Discrimination*	*Community Norms* *Local Structural Discrimination*	*Societal Norms* *Societal Structural Discrimination* Racism Classism Sexism Homophobia/Transphobia
**Health Care System**	*Insurance Coverage* Mi Salud program Private insurance Medicare/Medicaid *Health Literacy* *Treatment Preferences* Focus on remedial instead of prevention medicine	*Patient-Clinical Relationship* *Medical Decision Making*	*Availability of Services* Migration of physicians Private practices Difficultly receiving referrals Community health centers *Safety Net Services* Community Health Centers	*Quality of Care* Health professional shortage Poorly coordinated care Long wait times *Health Care Policies* Department of Health Law (1912)
**Health Outcomes**	**Individual Health**	**Family/Organizational Health**	**Community Health**	**Population Health**

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
