# Peer review of "Health Disparities Research Framework Adaptation to Reflect Puerto Rico’s Socio-Cultural Context"

_ijerph, 2020, doi:10.3390/ijerph17228544_

Round 1

Reviewer 1 Report

The topic is interesting and its study necessary, but the article requires a review before being published. The structure of the paragraphs should be improved, as some of them intersperse data and aspects of other sections.

The bibliography concerning the social determinants of health should be named already in the introduction of the article, since this is a fundamental axis to justify the interest and objectives of the article.

Likewise, the references section itself should be revised in format and check that all the references cited in the text appear later in the bibliography.

The article is based on the description of statistical data, so it would be necessary to unify the dates to which these data refer. Sometimes we talk about current data and others of 2017. This makes it difficult to interpret the article.

At the methodological level, a more rigorous description of the data and the sources from which they are obtained would be necessary. Greater systematization of the presentation and analysis process.

Author Response

1) The concept social determinants of health was mentioned in the introduction as suggested.

2) The references section was revised.

3) We used the latest data available regarding the topics, sometimes we do not have current data regarding health conditions in Puerto Rico

4) Since the article is an essay which seeks to present an overview and reflect on the topic we did not seek to describe the data in great detail.

Reviewer 2 Report

The aim of the Essay "Health Disparities Research Framework Adaptation to Reflect Puerto Rico’s Socio-Cultural Context" is to describe how Puerto Rico’s living conditions influence adverse health outcome at an individual, community and population level using the NIMHD Research Framework, which considers multiple factors and their intersecting influence.

The manuscript is interesting and covers important issues. However, some minor changes should be done before publication:

- In the Introduction section, on page 2, after line 45 and before describing the aim, a brief summary of adverse health outcomes in Puerto Rico at the individual, community and population level should be included.

- On page 3, the text on lines 117-118 should be revised as it is not well understood.

- All the text should be reviewed and it should be verified that the journal's style rules are strictly followed as some errors have been detected. For example, on page 3 line 134 the following text appears: “Center a reduction of $ 4.3 million [18] (Colón Dávila, 2020).” The text (Colón Dávila, 2020) should be deleted. And something similar happens in line 138 with the text "disabilities [9] (CNE, 2019).”

Author Response

1) We provided a very brief summary of the adverse health outcomes in the introduction.

2) We looked at the text on lines 117-118 and found that is was clear and also part of it is a direct citation so it should not be changed.

3) We looked at the text to ensure that errors like the one mentioned were eliminated.

Reviewer 3 Report

It is urgent appeal for reducing health disparities in Puerto Rico, in which authors reviewed the history of social development and related health risks that people are facing and provided their consideration and suggestions. The following issues should be further considered if possible.

  1. About health disparities in Puerto Rico, authors just presented some examples; if possible, it will be better if author could categorize main health issues to several topics with data, for example, for adults, for children, for women, for elders, or for infectious and non-communicable diseases…
  2. About NIMHD Framework considering the Puerto Rican population, authors seemed not to present this frame very clearly but only focused on disparities of political governance. It will better if they could present concert contents included in this frame in text. Supplementary table seemed not be explained fully in text.

Author Response

1) We decided not to categorize main health issues by age, gender, etc. because it is difficult to find current data regarding health conditions in Puerto Rico desegregated according to demographic characteristics such as the ones mentioned before.

2) We added a paragraph discussing the framework and how it relates to the topics discussed before in the article.

Round 2

Reviewer 3 Report

Authors have addressed my comments, I have no addtional comments.